# Antimicrobial Properties of New Polyamines Conjugated with Oxygen-Containing Aromatic Functional Groups

**DOI:** 10.3390/molecules28227678

**Published:** 2023-11-20

**Authors:** Mario Inclán, Neus Torres Hernández, Alejandro Martínez Serra, Gonzalo Torrijos Jabón, Salvador Blasco, Cecilia Andreu, Marcel lí del Olmo, Beatriz Jávega, José-Enrique O’Connor, Enrique García-España

**Affiliations:** 1Institute of Molecular Science, University of Valencia, 46980 Valencia, Spain; neustorres23@gmail.com (N.T.H.); alejandro.martinez-serra@uv.es (A.M.S.); salvador.blasco@uv.es (S.B.); enrique.garcia-es@uv.es (E.G.-E.); 2Escuela Superior de Ingeniería, Ciencia y Tecnología, International University of Valencia—VIU, 46002 Valencia, Spain; 3Departament de Bioquímica i Biologia Molecular, Facultat de Biologia, University of Valencia, 46100 Valencia, Spain; gonzalotj19@hotmail.com (G.T.J.); m.del.olmo@uv.es (M.l.d.O.); 4Departament de Química Orgànica, Facultat de Farmàcia, University of Valencia, 46100 Valencia, Spain; 5Laboratory of Cytomics, Joint Research Unit CIPF-UVEG, Department of Biochemistry and Molecular Biology, Faculty of Medicine, University of Valencia, 46010 Valencia, Spain; beatriz.javega@uv.es (B.J.); jose.e.oconnor@uv.es (J.-E.O.)

**Keywords:** polyamine, antimicrobial, bacteria, yeast, synergy, bactericidal, fungistatic

## Abstract

Antibiotic resistance is now a first-order health problem, which makes the development of new families of antimicrobials imperative. These compounds should ideally be inexpensive, readily available, highly active, and non-toxic. Here, we present the results of our investigation regarding the antimicrobial activity of a series of natural and synthetic polyamines with different architectures (linear, tripodal, and macrocyclic) and their derivatives with the oxygen-containing aromatic functional groups 1,3-benzodioxol, ortho/para phenol, or 2,3-dihydrobenzofuran. The new compounds were prepared through an inexpensive process, and their activity was tested against selected strains of yeast, as well as Gram-positive and Gram-negative bacteria. In all cases, the conjugated derivatives showed antimicrobial activity higher than the unsubstituted polyamines. Several factors, such as the overall charge at physiological pH, lipophilicity, and the topology of the polyamine scaffold were relevant to their activity. The nature of the lipophilic moiety was also a determinant of human cell toxicity. The lead compounds were found to be bactericidal and fungistatic, and they were synergic with the commercial antifungals fluconazole, cycloheximide, and amphotericin B against the yeast strains tested.

## 1. Introduction

Antimicrobial resistance (AMR) has become a major global health problem. The estimates in 2016 painted a grim picture noting that AMR would become the leading cause of death, with a likelihood of 10 million deaths per year by 2050 [1]. However, other studies published in 2022 were more pessimistic about the situation, demonstrating that the previous prediction was probably an underestimation and that the mortality rate associated with AMR is increasing at a worrying pace: There were already close to 5 million deaths associated with bacterial AMR in 2019 [2]. The classical solution against AMR consists of developing new families of antimicrobials. However, there are several drawbacks for many of the compounds currently under clinical trials, such as their high cost of production, which is related to the complexity of their synthesis, and, in the case of natural products, the scarcity of sources [3].

Polyamines are simple organic molecules with two or more amine groups positively charged at physiological pH. Biogenic polyamines, such as putrescine, spermidine, and spermine, are widely distributed in all organisms in which they develop multiple cellular functions. Their metabolic synthesis and concentration are highly regulated due to their role in cell proliferation and apoptotic processes. Polyamines are also involved in several signalling pathways, can act as neuromodulators, and influence the properties of several neurotransmitter pathways involved in mental disorders [4]. The multiple activities induced by polyamines as a result of their interaction with different biological targets support the proposal highlighting the polyamine skeleton as a universal template model in which appropriate structural modifications would provide selectivity on specific processes [5,6].

Exogenous polyamines are efficiently absorbed by transport systems, and as endogenous ones, they can also affect different biological targets. In recent decades, polyamine research has become an important field for drug development, and their potential application as anticancer and antiproliferative agents, agonist–antagonist receptor ligands, or antiparasitic compounds, among others, has been studied [4]. The activity of novel polyamines against Gram-positive and Gram-negative bacteria has been analysed by different research groups, and they have found promising compounds with antibacterial and/or antibiofilm activity and sometimes with synergic effects on other known antimicrobial compounds. The positive charges of the amino groups have been considered a key factor in their antimicrobial activity [7,8,9,10,11]. Conjugated polyamines with lipophilic organic molecules have also been found effective in this context; in these cases, the structure and characteristics of the organic moiety play an important role in antimicrobial activities [12,13,14,15,16].

1,3-Benzodioxole is a naturally occurring organic group found in many plant-based food products, such as vanilla (piperonal), black pepper (piperine), and cinnamon (safrole), among others. More importantly, its aldehyde (2H-1,3-benzodioxole-5-carbaldehyde) is mass-produced, cheap, and readily available since it is widely used by the industry developing artificial flavours and perfumes. 1,3-benzodioxole is also exploited in the pharmaceutical industry, and it can be found as a component of many bioactive molecules. Among the pharmaceuticals bearing this unit, there are examples in the literature highlighting their use in anticancer drugs [17], HIV treatments [18], antiparkinsonian agents [19], anti-inflammatory or analgesic products [20,21], and anticonvulsant remedies [22]. It is also present in a natural pesticide, acting both as an algicide and a herbicide [23]. Another group of bioactive compounds bearing this moiety is psychoactive and stimulant drugs [24,25]. There are also examples of its derivatives that show antimicrobial activity [26,27,28,29,30].

In this context, with the purpose of developing new antimicrobial compounds, we set out to study the activities of various polyamines containing the 1,3-benzodioxole moiety. The polyamine core varied in the number of amine groups and its topology (linear, tripodal, and macrocyclic). Therefore, we determined the antimicrobial activity against bacteria (*Escherichia coli* and *Staphylococcus aureus*) and yeast. The possible synergies between them and commercial antimicrobials and their toxicity against a human cell line were also determined. These results allowed us to conclude that the topology of the polyamine scaffold was a determinant of the activity. We then extended the study to three other derivatives in which we conjugated the most promising polyamine macrocyclic core with three alternative oxygen-containing aromatic groups. It is also worth mentioning that all these compounds were prepared in good yields through a simple one-pot synthetic method, which is robust and easily scalable.

## 2. Results and Discussion

### 2.1. Synthesis

The structures of the first polyamines used in this work are shown in Figure 1. Considering these compounds, **1b**–**3b** were commercially purchased. The synthesis of **4b** and **5b** has been optimised in our laboratory, and the compounds are systematically prepared [31,32,33,34]. The conjugated derivatives **1a**–**5a** were obtained in the multigram scale in one-pot reactions and under mild conditions.

For the conjugation of the aromatic moiety, selective condensation between 3,4-methylenedioxybenzaldehyde (commonly known as piperonal or heliotropin) and the primary amino groups of the corresponding polyamine (**1b**–**5b**) in the proper stoichiometry was carried out. Then, the intermediate imine was reduced in situ with NaBH_4_ (Figure 1). The purification to obtain spectroscopically pure compounds was easily achieved via simple precipitation and the recrystallisation of the hydrochloride salts.

To our knowledge, the synthesis of compounds **1a**–**5a** is reported here for the first time, except for **3a**, which was described before by Sumoto’s group [26], but with a different procedure to the one reported here. The ^1^H, ^13^C, and ATR-IR spectra of the new compounds, as well as the X-ray diffraction structure of compound **3a**, can be found in the Electronic Appendix A.

### 2.2. Basicity of the Receptors

The capacity of a given compound to interact with its biomolecular target depends heavily on factors such as the overall charge of the molecule and the number of hydrogen bond donor/acceptor atoms. Since the polyamines presented here are polyprotic systems, understanding which species are present in the solution at the experimental pH values is essential. For this reason, potentiometric titrations in water were carried out.

The stepwise protonation constants obtained are shown in Table 1. As expected, in the studied pH range (2.5 < pH < 10.5), the compounds display as many protonation constants as the secondary amino groups present in each molecule. The protonation reactions of the central tertiary nitrogen atoms (in the case of **3a**, **4a**, and **5a**) or the pyridine nitrogen (in **5a**) were not detected since they tend to occur at a pH that falls outside the lower range of the analysis technique. This was expected considering the known lower basicity of tertiary amines in water and the electrostatic repulsion of the surrounding ammonium groups [35].

As anticipated, the overall basicity correlates with the number of secondary amino groups, with **4a** having the higher value, in terms of its six amino groups. Compounds **1a**, **3a**, and **5a**, with three secondary amino groups in their structures, have similarly low-log β values. By contrast, **2a**, with four secondary amino groups, shows an intermediate basicity. These protonation constants allowed us to build the corresponding distribution diagrams (Appendix A) and to calculate the percentage of species and the average number of positive charges (n) at pH 7.4 (last row in Table 1).

### 2.3. Determination of the Antimicrobial Activity

The antimicrobial activity of the polyamines was determined using a Gram-negative bacteria (*Escherichia coli* JM101), a Gram-positive bacteria (*Staphylococcus aureus* BH1), and two strains of the yeast *Saccharomyces cerevisiae* (laboratory strain BY4741 carrying the YEplac195 plasmid or its derived one YEplac195-PDR5, named PDR5 from now on). This plasmid allows for the overexpression of the gene encoding Pdr5, an ATP-binding cassette (ABC) transporter, which can export toxic substances out of the cell (see [36] and references therein). For comparison purposes, commercial antibacterial (doxycycline, gentamicin, ciprofloxacin, and ampicillin) and antifungal (fluconazole, cycloheximide, and amphotericin B) agents were also included as control in our analyses. The organic compound 1,3-benzodioxole was also considered with both kinds of microorganisms. To obtain information about their antimicrobial activities, experiments were carried out to determine the minimal inhibitory concentration (MIC), the minimal microbicidal concentration (MMC), and possible synergies with the control substances.

#### 2.3.1. Determination of the Minimal Inhibitory Concentration (MIC)

The results obtained for the MIC are shown in Table 2. The most relevant observation regarding the antibacterial MIC values was the important differences between the unsubstituted polyamines **1b**–**4b** and the ones combined with 1,3-benzodioxole **1a**–**4a**. Only polyamine **5b** had, in some cases, a similar or better behaviour than the conjugated compounds. This fact, together with the higher MIC value measured for the lowest-molecular-mass compound 1,3-benzodioxole seems to indicate that the activity in the conjugated derivatives is mainly due to the polyamine moiety. It is also worth mentioning that for the biogenic polyamines spermidine (**1b**) and spermine (**2b**), the obtained results agree with the data found in the literature about their antimicrobial activity. Based on these data and the results presented in this study, spermidine **1b** and its conjugated derivative **1a** showed MIC values similar to or higher than spermine **2b** and its derivative **2a** against *E. coli* [37], *S. aureus*, and the yeast strains. Considering the conjugated polyamines, the best antibacterial activity was found for **4a** and **5a**, followed by **3a** and **2a**. Polyamine **5b** was also very effective against *E. coli*. 

In yeast, like in bacteria, conjugated compounds **4a** and **5a** and polyamine **5b** showed the best results. For the commercial drug fluconazole, the susceptibility of the strain overexpressing Pdr5 protein was much higher due to its involvement in the efflux of this drug [38]. However, in the case of the new compounds considered here, no differences between the two yeast strains were found, which demonstrates that Pdr5 is not involved in their efflux.

The fact that, in most cases, the antimicrobial activity was much higher in the conjugated polyamines containing 1,3-benzodioxole than in each one of the pair of components seems to indicate that their association improves some properties, probably related to the increased lipophilicity provided by the 1,3-benzodioxole fraction. Only the results of polyamine **5b** are comparable (and even better especially in yeasts) to those obtained with conjugated polyamines. This could also be explained by considering that it is the most lipophilic polyamine, as deduced from its ClogD value (see the last column in Table 2, in which ClogD values at physiological pH are indicated).

According to their topology, the studied polyamines can be classified as linear (**1** and **2**), tripodal (**3** and **4**), and macrocyclic (**5**). Moreover, regarding the whole charge that they have at physiological conditions (pH = 7.4), we found that compounds **1a**, **3a**, and **5a** have between +2.1 and +2.7 charges; **2a** has +3.4; and **4a** has +4.5. In the case of both tripodal compounds (**3a** and **4a**), or both linear ones (**1a** and **2a**), the activity was higher in the derivative with an upper charge. However, the main factor determining the activity seems to be the topology, because comparing the three compounds with similar charge (**1a**, **3a**, and **5a**), higher activity is shown (except for *S. aureus*) by macrocyclic compound **5a** followed by the tripodal one **3a** (see Figure 2). Moreover, it was similar or higher for macrocyclic derivative **5a,** with a whole charge of +2.4, than for the tripodal **4a** and the linear **2a** with charges of +4.5 and +3.4, respectively.

Figure 3 represents the MIC value for the conjugated polyamines in each microorganism in relation to its ClogD. Again, the core polyamine structure provides a better explanation for the activity against bacteria and yeast than their lipophilia. For instance, **5a** had a ClogD value between **1a** and **2a** but showed greater antimicrobial activity than both. Moreover, the behaviour of **4a** and **5a**, as well as the pure polyamine **5b** (except for *S. aureus*), was similar despite the differences in lipophilia. In summary, the presence of a lipophilic component in the drug seems necessary to guarantee the accessibility to the membrane of the microorganisms, but once this is achieved, the most important factor that determines its activity is the topology of the polyamine.

#### 2.3.2. Determination of the Minimal Microbicidal (Bactericidal/Fungicidal) Concentration (MBC or MFC)

The determination of MBC/MFC was carried out for those compounds with superior antimicrobial activity. The results revealed that those displaying antibacterial activity were bactericides for both *E. coli* and *S. aureus* strains (the MBC values are indicated in Table 3). In the case of *S cerevisiae*, compounds **4a**, **5a**, and **5b** were found to be fungistatic: Growth was observed after inoculation on SC-ura plates of an aliquot of a sample resulting from the overnight incubation of yeast cells with each one of these compounds at a concentration of four times the MIC value. The representative images of these results are shown in Appendix A.

#### 2.3.3. Determination of the Synergies between Compounds and Control Antifungal and Antibacterial Substances Using the Checkerboard Titration Approach

The synergy effect among different drugs becomes evident when the overall therapeutic action of their combination is greater than the sum of effects caused by each individual component [39]. Synergistic combinations facilitate an increase in beneficial results because the use of lower doses of each constituent can reduce the side effects and toxicity associated with high quantities of individual drugs. Combination therapies are widely used for the treatment of the most severe diseases, such as cancer or AIDS [40] (see also the references therein).

To determine the possibility of synergy between the lead compounds and the commercial antimicrobials essayed, each one was tested using the checkerboard titration method [41,42]. For the interpretation of the results, the criteria of Loewe’s additive theory were followed, and for this purpose, the fractional inhibitory concentration index (FICI) was calculated, as described in the Materials and Methods section. In the case of bacteria, the experiments were conducted between each active polyamine and every one of the commercial antibacterial drugs indicated in Table 1; we only found synergy between **5b** and ciprofloxacin in *E. coli*. Conversely, for yeast strains, positive results were found between compounds **4a** and **5a** and the known antifungal agent amphotericin B, as well as between **4a**, **5a**, and **5b** and cycloheximide. Additionally, in the BY4741 transformed with the YEplac195 control plasmid, **4a**, **5a**, and **5b** were found to be synergic with fluconazole (Table 4). Figure 4 shows the representative images of the results obtained after growth in the plate of aliquots corresponding to the most meaningful tubes prepared for the checkerboard titration approach.

### 2.4. Compound Cytotoxicity on Human Cells

Table 5 and Appendix A show the results of the viability assay carried out to determine the cytotoxicity of the conjugated compounds **1a**–**5a** and the pure polyamine **5b** on the human Jurkat cell line. As can be seen in Table 5, the most polar one, compound **5b** (ClogD −5.36), showed the lowest cytotoxicity (IC_50_ 1252 μg/mL), while the most lipophilic one, compound **3a** (ClogD −1.15), was clearly the most cytotoxic (IC_50_ 1.86 μg/mL). The remaining compounds showed intermediate cytotoxicity, with two different ranges of values, one for linear compounds **1a** and **2a** and the second one for compounds **4a** and **5a**. Figure 5 shows the IC_50_ values (black column) for each compound, together with the corresponding MIC values for the microorganisms considered (light blue, dark blue, and grey columns). Considering all these data together, macrocyclic polyamine **5b** is the one that showed the highest therapeutic index in all cases. The conjugated polyamines **4a** and **5a** also presented acceptable values in *E. coli*, in contrast to the other compounds (**1a**, **2a**, and **3a**), which were found to be very toxic to human cells.

### 2.5. Antimicrobial Potential of Other Derived Compounds from Polyamine ***5b***

Although the benzodioxole fragment is a component of some drugs, its inhibitory effect on the activity of CYP (cytochrome P450)-dependent enzymes has been described in mammals and other species [43] (see also the references therein). For this reason, and in order to find the active compounds in which toxicity could be lower than in the conjugated polyamines described here thus far, we extended the study to three other polyamines based in the macrocyclic scaffold conjugated with other groups different from benzodioxole but with similar physicochemical characteristics and properties. Figure 6 shows the structure of these compounds (**6a**, **7a**, and **8a**) in which polyamine **5b** was condensed, as described in Figure 1, with 2-hydroxybenzaldehyde, 4-hydroxybenzaldehyde, and 2,3-dihydrobenzofuran-5-carbaldehyde, respectively. The synthesis of compounds **6a** and **7a** was recently reported by some of us [44], whereas compound **8a** is reported here for the first time.

As with the other compounds, the acid-base behaviour of these new derivatives was studied via potentiometric titrations in water. As for **5a** and **5b**, the protonation of the central tertiary nitrogen atoms and the pyridine ones was not detected. The percentage of species and the average number of positive charges, n, at pH 7.4 were calculated (Table 6).

The susceptibility of the microorganisms to these compounds was measured by determining the minimum inhibitory concentration (MIC) and the minimum microbicidal concentration (MMC) (Table 7 and Table 8). To assess whether the antimicrobial activity of the tested compounds was relevant, it was compared with the value determined previously for **5a** and **5b**. It could be observed that **6a** exhibited the highest antimicrobial activity for yeast and *E. coli*. In the case of *S. aureus*, it was similar to that for **5a**. On the other hand, **7a** and **8a** yielded results like those of **5a** and **5b** only in the case of *E. coli* and better than **5b** in *S. aureus*.

Next, we investigated possible interactions with commercial antimicrobials using the checkerboard assay. The results of these experiments, displayed in Table 9, showed no synergy in the case of bacteria with **6a**, but positive results were found with **7a** and gentamicin, as well as with **8a** and ampicillin, doxycycline, and gentamicin in *E. coli*. In the case of yeast strains, **6a** yielded positive results with cycloheximide and amphotericin B. Experiments were not carried out in those cases in which the MIC values were very high.

These results were confirmed by plating samples corresponding to the incubation of the microorganisms with concentrations corresponding to half of the MIC of every one of the antimicrobials and with a mixture of one-quarter of the MIC of both. 

To understand the level of safety in human cells, the cytotoxicity of polyamines **6a**, **7a**, and **8a** was determined using the Jurkat cell line of human lymphocytic leukaemia (ATCC TIB-152). The data obtained are shown in Table 10 and Figure 7, as well as Appendix A.

As observed, **6a** and **8a**, and especially **7a**, were found to be less toxic than **5a**. This demonstrates that the substitution of the benzodioxole group by other aromatic groups with similar physicochemical properties (e.g., size and lipophilicity) has a significant and positive impact. This fact may be attributed to the inhibitory effect of 1,3-benzodioxole on CYP-dependent enzymes, as mentioned above and demonstrated with other drugs, although further experiments are necessary to confirm this [28].

In all cases, and particularly against the bacterium *E. coli*, the increase in the therapeutic index points to the potential utility of all these compounds. It is important to note that although **5b** and **7a** may appear to be less suitable based on their higher MIC values, when considering the whole data, they appear to be very promising candidates.

### 2.6. Intracellular ATP Concentrations and Microbiolytic Activity

To gain further insight into the effect of the compounds described in this work that have relevant antimicrobial activity and contain the same polyamine scaffold, we carried out two additional experiments with compounds **5a** and **6a**: one for the determination of the intracellular ATP values and the other for the analysis of their capacity to induce cell lysis.

#### 2.6.1. Analysis of the Intracellular ATP Concentration in Cells Treated with the Most Active Compounds Considered in This Work

Lehtinen et al. [45] described the use of bioluminescence-based techniques for measuring bacterial viability. Since ATP can be produced only in catabolically active cells, in the absence of external ATP, bioluminescence generated in an ATP-dependent reaction provides information about cell metabolic conditions.

Following the procedure described in the Materials and Methods section, luciferase activity was used to determine the intracellular ATP concentration in the bacterial and yeast cells treated with polyamines **5a** and **6a**. Studies were carried out in parallel with other compounds with antimicrobial activity (ampicillin and doxycycline in bacterial strains; fluconazole, amphotericin B, and SDS in the case of yeasts).

According to the results shown in Figure 8, in yeast cells treated with a concentration 50-fold the MIC value of amphotericin B, a dramatic decrease in ATP intracellular levels was observed after overnight treatment. Under the same conditions, treatment with **5a** and **6a** resulted in higher ATP levels, which were similar to the ones measured in the presence of SDS at a concentration of 0.1% (*w*/*v*). In these cases, ATP amounts were reduced to approximately 50–70% relative to untreated cells. The effect of fluconazole in the depletion of nucleotide intracellular levels was very reduced, although it is worth mentioning that a concentration 30-fold the MIC value was used due to its lower solubility in aqueous solutions. 

Regarding *E. coli* cells, the effect of all the compounds at five-fold MIC concentrations was a reduction in the ATP intracellular levels to 40% or less. This decrease was especially relevant in the case of the treatments with ampicillin and **5a**, followed by doxycycline and **6a**. For *S. aureus*, similar results were obtained, although in this case, **6a** displayed a higher effect than **5a** and doxycycline. These results confirm the important effect of **5a** and **6a** on cell metabolic activity in bacterial cells.

#### 2.6.2. Determination of the Microbiolytic Activity of the Microbicidal Compounds Considered in This Work Using Time-Kill Kinetics

Although the precise mechanism of action of antibacterial polyamines has not been fully elucidated, the similarity found between the structural requirements for them and the known antimicrobial peptides (AMPs) seems to indicate that both kinds of compounds share the same pharmacophore model (the presence of several positive charges and hydrophobic fragments) [46]. This suggests that, at least in bacteria, both could have a similar mechanism of action, in which the negatively charged bacterial membrane is the primary target, and the electrostatic interaction between this membrane and the positively charged polyamine backbone is the major driving force that finally leads to the disruption of membrane integrity [15,16].

The experiments carried out with the studied polyamines revealed bactericidal behaviour for most of them (Table 3 and Table 8). In order to determine if these compounds were capable of inducing cell lysis in bacteria, time-kill assays were performed, following the procedure described in the Materials and Methods section. For these experiments, compounds **5a** and **6a** were used again. Cell growth was followed by the OD_600_ measurement in the absence and presence of polyamines of interest and also using control agents with known lytic effects (ampicillin) [47] and non-lytic properties (doxycycline) [48] (see also references therein) (Figure 9). 

In the case of *E. coli*, when cells were treated with ampicillin, and in accordance with its lytic activity, growth was not observed over time; in fact, OD_600_ significantly decreased as incubation time increased. The obtained results for **5a** and **6a** demonstrate that they do not have a lytic effect on this bacterium and behave similarly to doxycycline. Regarding *S. aureus*, the ampicillin treatment results were also obtained for cellular lysis, which was not the case for doxycycline and the polyamines considered in this study. However, with the methodology used in these experiments, some degree of effect on membrane integrity cannot be completely ruled out.

Experiments were also carried out with yeast cells and, as expected considering its fungistatic character, the results demonstrate that these compounds do not induce lysis, despite what occurred with amphotericin B.

## 3. Materials and Methods

### 3.1. Synthesis

All reagents and chemicals were obtained from commercial sources and were used as received. The solvents employed for the chemical synthesis were of analytical grade and were utilised without further purification. Polyamines **1b**–**3b** were commercially purchased, whereas **4b**, **5b, 6a**, and **7a** were prepared following procedures previously described [24,32,33,34,44]. All compounds were characterised via ^1^H and ^13^C NMR spectroscopy and elemental analysis.

For the synthesis of conjugated polyamines **1a**–**8a**, the corresponding polyamine (between 0.5 and 1 g) was first dissolved in 30–60 mL of methanol and placed in a 250 mL round flask with magnetic stirring. The number of equivalents of each aldehyde (2H-1,3-benzodioxole-5-carbaldehyde, 2-hydroxybenzaldehyde, 4-hydroxybenzaldehyde, or 2,3-dihydrobenzofuran-5-carbaldehyde) corresponding to the number of primary amines was dissolved in another 30 mL of methanol and added dropwise to the reaction flask. The temperature was then slowly increased to 50 °C and maintained under magnetic stirring for 24 h. After this time, it was lowered to 0 °C, and an excess of NaBH_4_ was slowly added (8 equivalents × No. of imines). The solution was continuously stirred until it reached room temperature, and then the solvent was eliminated. The crude product of the reaction was dissolved in 100 mL of water and extracted four times with 25 mL of dichloromethane. The product was recrystallised by adding an excess of HCl 4 M in dioxane, centrifuged, and dried under vacuum (60–90% yields).

*N^1^-(benzo[d][1,3]dioxol-5-ylmethyl)-N^4^-(3-((benzo[d][1,3]dioxol-5-ylmethyl)amino)propyl)butane-1,4-diamine* (**1a**). ^1^H RMN (D_2_O, 300 MHz): δ_H_ 6.88 (m, 3H); 5.91 (s, 6H); 4.09 (s, 6H); 3.05 (m, J = 5, 6H); 2,03 (m, 6H); 1.68 (m, 6H). ^13^C RMN (D_2_O, 75 MHz): δ_C_ 148.8, 148.1, 124.2, 109.9, 108.9, 101.6, 51.0, 46.9, 45.5, 44.1, 43.1, 22.1. Anal. Calc. for C_23_H_34_N_3_O_4_Cl_3_: C, 52.8; H, 6.6; N, 8.0; O, 12.2; Cl, 20.4. Exp.: C, 51.9; H, 6.3; N, 7.2; O, 12.6; Cl, 22.0.

*N^1^,N^1′^-(butane-1,4-diyl)bis(N^3^-(benzo[d][1,3]dioxol-5-ylmethyl)propane-1,3-diamine)* (**2a**). ^1^H RMN (D_2_O, 300 MHz): δ_H_ 6.88 (m, 3H); 5.92 (s, 6H); 4.11 (s, 6H); 3.06 (m, 6H); 2,03 (m, 6H); 1.70 (m, 6H). ^13^C RMN (D_2_O, 75 MHz): δ_C_ 148.3, 147.7, 124.2, 123.9, 118.1, 109.9, 108.9, 101.6, 51.1, 47.0, 44.4, 43.6, 22.7. Anal. Calc. para C_26_H_42_N_4_O_4_Cl_4_·H_2_O: C, 49.9; H, 6.9; N, 9.0; O, 11.5; Cl, 22.7. Exp.: C, 50.0; H, 6.7; N, 8.3; O, 11.1; Cl, 23.9.

*N^1^-(benzo[d][1,3]dioxol-5-ylmethyl)-N^2^,N^2^-bis(2-((benzo[d][1,3]dioxol-5-ylmethyl)amino)ethyl)ethane-1,2-diamine* (**3a**) ^1^H RMN (D_2_O, 300 MHz): δ_H_ 6.86 (m,10H); 5.92 (s, 6H); 4.06 (s, 6H); 3.04 (m, J = 6, 6H); 2.86 (t, J = 6, 5H). ^13^C RMN (D_2_O, 75 MHz): δ_C_ 148.2, 147.7, 124.3, 123.9, 109.5, 109.1, 101.6, 51.0, 49.1, 44.7, 44.5, 43.8, 22.4. Anal. Calc. for C_30_H_39_N_4_O_6_Cl_3_(H_2_O): C, 53.3; H, 6.11; N, 8.29. Exp.: C, 53.0; H, 8.7; N, 8.1.

*N^1^-(benzo[d][1,3]dioxol-5-ylmethyl)-N^3^-(2-((2-((3-((benzo[d][1,3]dioxol-5-ylmethyl)amino)propyl)amino)ethyl)(2-((3-(((2,3-dihydrobenzofuran-6-yl)methyl)amino)propyl)amino)ethyl)amino)ethyl)propane-1,3-diamine* (**4a**). ^1^H RMN (D_2_O, 300 MHz): δ_H_ 6.86 (m,10H); 5.87 (s, 6H); 4.07 (s, 6H); 3.1 (m, J = 6, 16H); 2.79 (t, J = 6, 5H); 2.11 (m, 6H). ^13^C RMN (D_2_O, 75 MHz): δ_C_ 148.2, 147.7, 124.3, 123.9, 109.5, 109.1, 101.6, 51.0, 49.1, 44.7, 44.5, 43.8, 22.4. Anal. Calc. for C_39_H_63_N_7_O_6_Cl_6_(H_2_O): C, 49.90; H, 6.77; N, 10.45; O, 10.23. Exp.: C, 48.74; H, 6.63; N, 9.67; O, 11.73.

*2-(3,6,9-triaza-1(2,6)-pyridinacyclodecaphane-6-yl)-N-(benzo[d][1,3]dioxol-5-ylmethyl)ethan-1-amine* (**5a**). ^1^H NMR (D_2_O, 300 MHz): δ_H_ 7.97 (t, J = 7.80, 1H); 7.46 (d, J = 7.80, 2H); 6.99 (m, 3H); 6.03 (s, 2H); 4.64 (s, 4H); 4.21 (s, 2H); 3.28 (m, 6H); 3.06 (t, J = 5.3, 2H); 2.92 (t, J = 5.3, 4H). ^13^C NMR (D_2_O, 75 MHz): δ_C_ 148.96, 148.20, 147.68, 139.62, 124.42, 124.00, 122.02, 110.11, 106.91, 101.51, 51.12, 50.93, 50.43, 49.51, 45.84, 42.21. Anal. Calc. for C_21_H_32_N_5_O_2_Cl_3_(H_2_O): C, 49.37; H, 6.71; N, 13.71; O, 9.39. Exp.: C, 47.87; H, 6.33; N, 11.96; O, 9.60.

*2-(3,6,9-triaza-1(2,6)-pyridinacyclodecaphane-6-yl)-N-((2,3-dihydrobenzofuran-5-yl)methyl)ethan-1-amine* (**8a**). ^1^H NMR (D_2_O, 300 MHz): δ_H_ 7.89 (t, J = 7.80, 1H); 7.37 (d, J = 7.80, 2H); 7.30 (s, 1H); 7.18 (dd, J= 8.3, 1H); 6.79 (d, J = 8.3, 1H); 4.56 (m, 5H); 4.14 (s, 2H); 3.17 (m, 6H); 2.97 (m, 2H); 2.84 (t, J = 5.3, 4H). ^13^C NMR (D_2_O, 75 MHz): δ_C_ 160.06, 148.84, 139.73, 130.19, 128.96, 127.04, 122.77, 122.14, 109.51, 72.10, 66.53, 51.28, 50.85, 50.44, 49.42, 45.85, 42.07. Anal. Calc. for C_22_H_34_N_5_OCl_3_(H_2_O)_2_: C, 50.14; H, 7.27; N, 13.29; O, 9.11. Exp.: C, 49.69; H, 6.61; N, 12.58; O, 8.19.

### 3.2. Potentiometric Titrations

The potentiometric titrations were carried out at 298.1 ± 0.1 K using NaCl 0.15 M as a supporting electrolyte. The experimental procedure (burette, potentiometer, cell, stirrer, microcomputer, etc.) has been fully described elsewhere [49]. The acquisition of the EMF data was performed with the computer program PASAT [50]. The reference electrode was a Ag/AgCl electrode in a saturated KCl solution. The glass electrode was calibrated as a hydrogen-ion concentration probe via the titration of previously standardised amounts of HCl with CO_2_-free NaOH solutions and the equivalent point determined using the Gran method [51,52], which provides the standard potential, *E^θ^′*, and the ionic product of water (pK_w_ = 13.73(1)).

The computer program HYPERQUAD was used to calculate the protonation constants [53]. The pH range investigated was 2.5–11.0, and the concentration of the ligands ranged from 5 × 10^−4^ to 5 × 10^−3^ mol/dm. The different titration curves for each system (at least two) were treated either as a single set or as separated curves without significant variations in the values of the stability constants. Finally, the sets of data were merged and treated simultaneously to give the final stability constants.

### 3.3. Antimicrobial Activity

#### 3.3.1. Bacterial and Yeast Strains and Growth Conditions

The microorganisms used for the susceptibility tests were the bacteria *Escherichia coli* JM101 and *Staphylococcus aureus* BHI, and the yeast *Saccharomyces cerevisiae* strains BY4741 (*MAT a his3Δ1 leu2Δ0 met15Δ0 ura3Δ0* (EUROSCARF)) carrying YEplac195 (from EUROSCARF) or YEPlac195-PDR5 (provided by Ayse Banu Demir, Izmir University of Economics, Turkey) plasmids.

Bacterial strains were usually maintained and grown in an LB medium (0.5% (*w*/*v*) yeast extract, 1% (*w*/*v*) bactotryptone, and 1% (*w*/*v*) NaCl) at 37 °C. For MIC (minimal inhibitory concentration) and MMC (minimal microbicidal concentration) assays, and for the determination of synergies, overnight growth was carried out in 2% (*w*/*v*) bactopeptone (BP), and then the cultures were diluted and incubated for several hours in 2% (*w*/*v*) BP for ensuring logarithmic growth. Yeast strains were grown in an SC-ura medium (0.17% (*w*/*v*) yeast nitrogen base without amino acids and ammonium sulphate, 0.2% (*w*/*v*) drop-out without uracil, 0.5% (*w*/*v*) (NH_4_)_2_SO_4_, and 2% (*w*/*v*) glucose) at 30 °C. Liquid cultures were incubated with shaking (200 rpm). Solid media also contained 2% (*w*/*v*) agar.

#### 3.3.2. Preparation of Stock Solutions of Test and Control Compounds

The control drugs used in our experiments were the commercial antibacterial doxycycline, gentamicin, ciprofloxacin, and ampicillin, and the antifungal fluconazole, amphotericin B, and cycloheximide. All of them were purchased from Sigma-Aldrich (Darmstadt, Germany). Stock solutions of these drugs and the compounds to be tested were prepared in phosphate buffer 20 mM pH 7.4.

#### 3.3.3. Determination of the Minimum Inhibitory Concentration (MIC)

For the analysis of the antimicrobial activity, the strategies and indications of Cushnie et al. were considered [54]. The standardised method of microdilution in a culture medium was followed to determine the MIC [55]. In bacterial strains, serial dilutions of the compounds to be tested and the control antibiotics were prepared in phosphate buffer (20 mM, pH 7.4, 0.5 mL) and were mixed with a suspension of bacterial cells in a logarithmic growth phase (in 1% BP, 0.5 mL). The final OD_600_ of the whole solutions were 0.004 and 0.008 for *E. coli* and *S. aureus* cultures, respectively. After 20 h of incubation at 37 °C with shaking, the MIC was determined as the lowest concentration of each compound in which no visible bacterial growth was observed. 

The determination of the MIC for yeast strains was carried out following a similar procedure to that of the polyamines and antifungal reference agents: Serial dilutions of each one in phosphate buffer (20 mM, pH 7.4, 0.5 mL) were prepared and mixed with a suspension of yeast cells (OD_600_ 0.04 in SC-ura 2X (VWR, Atlanta, GA, USA), 0.5 mL). After 24 h of incubation at 30 °C with shaking, the MIC was determined as the lowest concentration of each compound that resulted in the same optical density of the solution as that observed prior to the incubation. Experiments were carried out at least three times in all cases.

#### 3.3.4. Determination of the Minimum Microbicidal Concentration (MMC)

Once the MIC was known, the samples were prepared as described in the previous section with polyamine concentrations corresponding to the MIC and half, double, and four times its value. After overnight incubation with the microorganisms, 30 μL of a ten-fold dilution of each one was spread in LB (in the case of bacterial strains) or SC-ura (for yeast strains) plates. They were incubated for 20 h at 37 °C (bacteria) or 48 h at 30 °C (yeasts), and the level of growth was determined. Since the microbiostatic activity has been defined as an MMC-to-MIC ratio of >4 [56], we classified compounds as microbiostatic if growth was observed in the plates from samples containing 4 times the MIC concentration, and in microbicides, those in which no growth was detected at this concentration were considered microbiostatic. In the case of microbicidal substances, the minimal fungicidal/bactericidal concentration (MFC/MBC) was determined as the lowest concentration of the tested agent that either completely prevented growth or resulted in a >99.9% decrease in the inoculum. Experiments were carried out at least three times.

#### 3.3.5. Determination of the Synergies between the Polyamines and Control Antifungal and Antibacterial Substances Using the Checkerboard Titration Approach

The combined antimicrobial activity of the test compounds with the antibacterial or antifungal control agents was performed under the same conditions used for the MIC determination and following the procedures described by Carton Herrán [57], Reis de Sá et al. [58] and Zharkova et al. [55]. For each experiment, 16 samples (4 rows × 4 columns) in a total volume of 0.5 mL were prepared in tubes using an array of combinations between the MIC value and 0 μg/mL of the compound to be tested in each row, and between the MIC value and 0 μg/mL of the reference drug in each column. In this way, a variety of mixtures with different concentrations of them alone or in combination was obtained. Briefly, 0.5 mL of a microorganism culture was added to each one of the samples, and after incubation for 24 h with shaking at the appropriate temperature (37 °C or 30 °C), the tubs were visually inspected, and the OD_600_ was measured. From this information, the fractional inhibitory concentration index (FICI) was calculated according to the formula *FIC index* = ([*A*]/[*CMI A*]) + ([*B*]/[*CMI B*]), where [*A*] and [*B*] are respective concentrations of tested polyamine and reference drug in their combination effectively inhibiting the growth of bacteria or yeast, and [*MIC A*] and [*MIC B*] are the individual MICs of the tested polyamine and reference drug when they are used alone. An interaction is considered synergistic when the FICI value is ≤0.5 [41,42].

To confirm the data obtained in the checkerboard assay, 30 μL of a ten-fold dilution of the tubes containing concentrations of half of the MIC of each one of the compounds alone (polyamine to be tested and reference drug) and one fourth of the MIC of each one in combination were also spread in LB or SC-ura plates (according to the microorganism considered in each case) and incubated at 37 °C or 30 °C. The visualisation of these plates was performed after 24 or 48 h, respectively. All these experiments were carried out in triplicate.

#### 3.3.6. Analysis of the Intracellular ATP Concentration in Bacterial and Yeast Cultures Treated with the Tested Compounds

The procedure described by Lehtinen et al. was followed [45], with the modifications indicated herein. In the case of bacteria, 1.5 OD_600_ units were incubated overnight in 1 mL of final volume with 5-fold MIC concentrations. Then, the cells were collected, washed 3 times in 20 mM phosphate buffer pH 7.4, and lysed in 200 μL of LS1 (50 mM of glucose, 25 mM of Tris-HCl pH 8.0, and 10 mM of EDTA pH 8.0) containing 5 mg/mL of lysozyme and the same volume of glass beads (425–600 μm), by stirring 10 times for 15 s each. Then, the solution was centrifuged at 14,800× *g* for 15 min, and the supernatant was used for a luciferase assay. For this purpose, in each well of a microtiter plate, 100 μL of luciferase assay buffer (0.015 M of MgSO_4_, 0.015 M of KH_2_PO_4_ pH 8, 0.004 M of EGTA pH 8, and 2 mM of DTT) was added together to a 50 μL of 0.3 mg/mL luciferin solution in 0.01% (*w*/*v*) Triton X-100 and 50 μL of the cell extract. Finally, 15 μL of a 67 μg/mL aqueous solution of luciferase was added. Light emission was immediately detected using a Luminoskan Ascent instrument (Thermo Scientific, Waltham, MA, USA) for 1 s per well. Glass beads, lysozyme, luciferin, and luciferase were purchased from Sigma-Aldrich.

In the case of yeasts, 50 OD_600_ units were incubated overnight in 1 mL of final volume with 50-fold MIC concentrations. Then, cells were collected and washed as above and resuspended in 500 μL of LS2 (140 mM of NaCl, 1.5 mM of MgCl_2_, 10 mM of Tris-HCl pH 8.0, and 0.5% (*v*/*v*) NP40). Cells were lysed in the presence of 1 volume of glass beads as described above. The next steps were carried out as in the case of bacterial samples.

Experiments were carried out at least in triplicate in all cases, and control drugs and SDS (0.1% (*w*/*v*) of the final concentration) were also included.

#### 3.3.7. Analysis of the Lytic Activity of Microbicidal Compounds

To determine if the compounds showing microbicidal activity induce cell lysis, the procedure described by Fleeman et al. was followed [59], with the modifications indicated below. The *E. coli* and *S. aureus* strains were grown as in the experiments described earlier. The cells corresponding to 1.5 units of OD_600_ were collected and resuspended in 1% BP after washing in the same medium. They were incubated in a final volume of 1 mL with concentrations 5 times higher than the MIC of the analysed compounds, and OD_600_ was monitored over time. Other antimicrobial compounds were also used in this study as controls. The experiments were performed in triplicate.

### 3.4. Determination of Compound Cytotoxicity on Human Cells

The cytotoxicity of the compounds was assessed via flow cytometry with a viability assay using propidium iodide (PI), a DNA-binding fluorescent probe not permeant to live cells, which is commonly used to detect dead cells in a population [60]. The human lymphocytic leukaemia Jurkat cell line (ATCC TIB-152) was obtained from the American Type Cell Culture (Sigma-Aldrich) and grown in suspension in an RPMI 1640 medium (Thermo Fisher Scientific, Waltham, MA, USA) supplemented with L-glutamine, antibiotics, and 10% heat-inactivated foetal calf serum (FCS, Thermo Fisher Scientific, Waltham, MA, USA). For the viability assay, cells were seeded in plastic 96-well plates at 250,000 cells/μL in an RPMI 1640 medium supplemented with 5% FCS. The cells were treated for 24 h with a range of concentrations of the test compounds or with the same volume of phosphate buffer (PBS) in control wells. After incubation, cell suspensions were washed, resuspended in 400 μL fresh medium, and incubated for 5 min at room temperature in the dark with 3.75 μM PI (Sigma-Aldrich). Cell viability was determined via flow cytometry using a Gallios flow cytometer (Beckman Coulter). PI fluorescence was excited with an argon-ion laser at 488 nm and collected through a 620/630 nm band-pass emission. Cellular events were discriminated from debris using forward scatter (FS) and side scatter (SS) signals. Cell aggregates were excluded from analysis based on FS-integral and FS-peak. The gates applied for population discrimination were set manually based on control samples. For each sample, 10,000 events were collected. For further analysis, the cytometric data were exported to Kaluza 2.1 software (Beckman Coulter, Brea, CA, USA). Dead cells were identified and quantified as PI-positive events. The cytotoxic potency of test compounds was quantified using their IC_50_ values, i.e., the compound concentration at which cell viability is inhibited by 50%, as calculated via curve fitting and determined with the correlation coefficient (R2) using GraphPad Prism 9.0 software. Experiments were carried out in triplicate.

## 4. Conclusions

In this work, we describe a collection of new compounds based on different polyamine scaffolds functionalised with different oxygen-containing aromatic groups that are easily synthesised and display antibacterial/antifungal activity. As an overall conclusion, it was demonstrated that the addition of a lipophilic moiety to the polyamine backbone enhances the efficacy of the resulting compounds. Structural characteristics such as total charge and the polyamine topology seem to be determinative in the antimicrobial activity. Moreover, these substances are interesting antifungal agents in combination with other known drugs, such as cycloheximide and amphotericin. Macrocyclic polyamine **5b**, its derivatives **5a** and **6a**, and tripodal compound **4a** were found to be the most effective antimicrobials. However, the nature of the lipophilic component was determinative in the toxicity of these derivatives on human cells, and in this context, polyamines **5b** and **7a** also proved promising as alternatives to lead due to their low harmfulness. It would be interesting to advance the research of similar compounds, designed through the rational modification of their structure that could be more selective and effective. Further analyses using computational, biochemical, and microscopy-based methods would also be required to understand their mechanism of action on cells.

## Data Availability

The data presented in this study are available upon request from the corresponding authors.

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
