# Peer review of "Antimicrobial Properties of New Polyamines Conjugated with Oxygen-Containing Aromatic Functional Groups"

_molecules, 2023, doi:10.3390/molecules28227678_

Round 1

Reviewer 1 Report

Comments and Suggestions for Authors

The study characterizes the antimicrobial and antifungal properties of polyamine derivatives. The authors synthesized chemical compounds and characterized the obtained products using NMR and elemental analysis. A number of studies were carried out on the biological activity of the obtained chemical compounds. The work has been thoroughly prepared, the charts and descriptions are clear and lucid. My only complaint/question concerns the analysis of measurement errors. How many repetitions were individual measurements performed? What is the measurement error (standard deviation). What is the repeatability of the results, which is important for this type of research. Were the measurements performed repeatedly? After dispelling these doubts, I recommend the article for publication in the Molecules journal.

Author Response

We want to thank the Reviewers for their kind and valuable comments and suggestions. All aspects have been considered in the revised version of the manuscript and the changes have been highlighted in the text in yellow (Reviewer 1), blue (Reviewer 2) and green (Reviewer 3), so they can be easily followed. We hope that the Reviewers will now find this new version suitable for publication.

Here are the point-by-point responses:

Reviewer 1

Comments and Suggestions for Authors

The study characterizes the antimicrobial and antifungal properties of polyamine derivatives. The authors synthesized chemical compounds and characterized the obtained products using NMR and elemental analysis. A number of studies were carried out on the biological activity of the obtained chemical compounds. The work has been thoroughly prepared, the charts and descriptions are clear and lucid. My only complaint/question concerns the analysis of measurement errors. How many repetitions were individual measurements performed? What is the measurement error (standard deviation). What is the repeatability of the results, which is important for this type of research. Were the measurements performed repeatedly? After dispelling these doubts, I recommend the article for publication in the Molecules journal.

Throughout the Material and Methods section and the Figures and Tables we have provided information about repeats and standard deviation (see legends of Figure 8 and 9; footnotes of Tables 1, 2, 5, 6, 7 and 10, and section 3.3 in Materials and Methods). On the other hand, we have added this information in Tables 3, 4, 8 and 9 and in section 3.4 in Materials and Methods (highlighted in yellow), where we did not include this data in the first version of the manuscript. In Figures 1 to 7, the inclusion of this information does not proceed.

Reviewer 2 Report

Comments and Suggestions for Authors

Title: Antimicrobial properties of new polyamines conjugated with 2 oxygen containing aromatic functional groups

Journal: Molecules

This manuscript describes an interesting study on the antimicrobial properties of new polyamines.

Below are some minor remarks to improve the manuscript:

·        Page 2, line 46: add definition or a short description of polyamines for readers.

·        Page 2, line 85: as this is the first mention of E. coli and S. aureus, it is necessary to write the whole names: Escherichia coli and Staphylococcus aureus and the yeast strain of Saccharomyces cerevisiae.

·        Page 4, chapter 2.3: as this is the first mention of the antimicrobial activity, it is necessary to add the strains of Escherichia coli and Staphylococcus aureus and to add Saccharomyces cerevisiae. I also propose not to use the term ‘two types of microorganisms’, this is a scientific article, not a lay text. Perhaps writing something like this: ‘we used a grampositive bacteria (Staphylococcus aureus BHI), a gram-negative bacteria (Escherichia coli JM101) and two strains of the yeast Saccharomyces cerevisiae (BY4741 carrying the YEplac195 plasmid or its derived one YEplac195-PDR5)’.

·        All tables and figures, where relevant: it makes sense to always use the same order of the chosen microorganisms for determining the antimicrobial effect. The strain information is not in italics.

Author Response

We want to thank the Reviewers for their kind and valuable comments and suggestions. All aspects have been considered in the revised version of the manuscript and the changes have been highlighted in the text in yellow (Reviewer 1), blue (Reviewer 2) and green (Reviewer 3), so they can be easily followed. We hope that the Reviewers will find this new version suitable for publication in this Journal.

Here are the point-by-point responses:

Reviewer 2

  • Page 2, line 46: add definition or a short description of polyamines for readers.

A definition of polyamines has been included at the beginning of the paragraph.

  • Page 2, line 85: as this is the first mention of E. coli and S. aureus, it is necessary to write the whole names: Escherichia coli and Staphylococcus aureus and the yeast strain of Saccharomyces cerevisiae.

These changes have been made according the reviewer suggestions (highlighted in blue color)

  • Page 4, chapter 2.3: as this is the first mention of the antimicrobial activity, it is necessary to add the strains of Escherichia coli and Staphylococcus aureus and to add Saccharomyces cerevisiae. I also propose not to use the term ‘two types of microorganisms’, this is a scientific article, not a lay text. Perhaps writing something like this: ‘we used a grampositive bacteria (Staphylococcus aureus BHI), a gram-negative bacteria (Escherichia coli JM101) and two strains of the yeast Saccharomyces cerevisiae (BY4741 carrying the YEplac195 plasmid or its derived one YEplac195-PDR5)’.

These changes have been made according the reviewer suggestions (highlighted in blue color)

  • All tables and figures, where relevant: it makes sense to always use the same order of the chosen microorganisms for determining the antimicrobial effect. The strain information is not in italics.

We have revised all Figures and Tables according to the suggestions of the reviewer. In all cases we have followed the order: E. coli, S. aureus and yeast. Modifications have been made in Figure 8 and Tables 2, 3, 7, 8 and 9. In Figure 5 we have also deleted a line in the Y-axis and modified a mistake in the legend (“5b” has been changed by “5a”).

Reviewer 3 Report

Comments and Suggestions for Authors

Dear authors of the manuscript

Antimicrobial properties of new polyamines conjugated with 2 oxygen containing aromatic functional groups think your experimental design and represented results are of good scientific outcome, but there are some corrections are needed

1-figure 1 legend should be improved 

2-please improve the english language and typographical errors

3- revise the abbreviations of the unites all over the manuscript and uniform it, ex: is it h. or h

4- in materials and method section , you mentioned you did H and C13 NMR, but the C13 nmr are not present in the main manuscript or the supplementary files, please ad the C13 nmr

5- add the FTIR charts or the Raman spectral data for the new compounds

6- in figure 5 and 7, please split each figure in two figures; one for the IC50 and the other for theMIC

7-the conclusion should be more specific for the represented results with future prospective

thank you

Comments on the Quality of English Language

the english need minor editing

Author Response

We want to thank the Reviewers for their kind and valuable comments and suggestions. All aspects have been considered in the revised version of the manuscript and the changes have been highlighted in the text in yellow (Reviewer 1), blue (Reviewer 2) and green (Reviewer 3), so they can be easily followed. We hope that the Reviewers will find this new version suitable for publication in this Journal.

Here are the point-by-point responses:

Reviewer 3

1-figure 1 legend should be improved 

The font size of the legend in figure 1 has been increased and the position of the numbers have been modified to improve its readability.

2-please improve the english language and typographical errors

The manuscript has been revised to correct typographical errors and improve some sentences.

3- revise the abbreviations of the unites all over the manuscript and uniform it, ex: is it h. or h

We have revised the manuscript, so all the units are coherent and uniform.

4- in materials and method section, you mentioned you did H and C13 NMR, but the C13 nmr are not present in the main manuscript or the supplementary files, please ad the C13 nmr

We thank the reviewer for the suggestion. The 13C-NMR spectra of all the new compounds have now been included in the Supplementary Materials as Figure S2.

5- add the FTIR charts or the Raman spectral data for the new compounds

As suggested by the reviewer, we have now measured and included the ATR-IR spectra of all the new compounds. They can be found in the Supplementary Material as Figure S3.

6- in figure 5 and 7, please split each figure in two figures; one for the IC50 and the other for theMIC

We thought that the introduction of both series of values jointly in a bar chart figure could help the reader to visually compare them and get a better understanding of the compounds with potential interest according to both criteria. Taking this into account, and considering that the other reviewers have not introduced any criticism regarding this point, we consider that the best option is to maintain both figures. The alternative would their deletion because the same data are shown in Tables 5 and 10.

7-the conclusion should be more specific for the represented results with future prospective

A series of modifications have been introduced in the Conclusions for them to be more specific and representative of the presented work. Specific experiments are now mentioned in the future perspective.

Round 2

Reviewer 1 Report

Comments and Suggestions for Authors

I accept it after the corrections

Reviewer 3 Report

Comments and Suggestions for Authors

no comments